# Analysis and Experiment of Laser Energy Distribution of Laser Wireless Power Transmission Based on a Powersphere Receiver

Tiefeng He [1], Guoliang Zheng [1], Qingyang Wu [2], Haixuan Huang [2], Lili Wan [1], Keyan Xu [1], Tianyu Shi [1] and Zhijian Lv [1,*]

---

1 Sino-German College of Intelligent Manufacturing, Shenzhen Technology University, Shenzhen 518118, China; hetiefeng@sztu.edu.cn (T.H.); zhengguoliang@sztu.edu.cn (G.Z.); wanlili@sztu.edu.cn (L.W.); xukeyan@sztu.edu.cn (K.X.); shitianyu@sztu.edu.cn (T.S.)
2 College of Big Data and Internet, Shenzhen Technology University, Shenzhen 518118, China; wuqingyang@sztu.edu.cn (Q.W.); huanghaixuan@sztu.edu.cn (H.H.)
* Correspondence: lvzhijian@sztu.edu.cn

**Abstract:** Laser wireless power transmission (WPT) is one of the most important technologies in the field of long-range power transfer. This technique uses a laser as a transmission medium instead of conventional physical or electrical connections to perform WPT. It has the characteristics of long transmission distance and flexible operation. The existing laser wireless power transmission system uses photovoltaic cells as a receiver, which convert light into electricity. Due to the contradiction between the Gaussian distribution of laser and the uniform illumination requirements of photovoltaic cells, the laser wireless power transmission technology has problems such as low transmission efficiency and small output power. Therefore, understanding the energy distribution changes in the laser during transmission, especially the energy change after the laser is transmitted to each key device, and analyzing the influencing factors of the energy distribution state, are of great significance in improving the transmission efficiency and reducing the energy loss in the system. This article utilizes the optical software Lighttools as a tool to establish a laser wireless power transmission model based on a powersphere. This model is used to study the energy distribution changes in the laser as it passes through various components, and to analyze the corresponding influencing factors. To further validate the simulation results, an experimental platform was constructed using a semiconductor laser, beam expander, Fresnel lens, and powersphere as components. A beam quality analyzer was used to measure and analyze the laser energy distribution of each component except for the powersphere. The output voltage and current values of various regions of the powersphere were measured using a multimeter. The energy distribution of the powersphere was reflected based on the linear relationship between photo-generated current, voltage, and light intensity. The experimental results obtained were in good agreement with the simulation results. Simulations and experiments have shown that using a beam expander can reduce divergence angle and energy loss, while employing large-aperture focusing lens can enhance energy collection and output power, providing a basis for improving the efficiency of laser wireless power transfer.

**Keywords:** laser; wireless energy transmission; conversion efficiency; model; powersphere

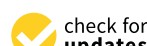



## 1. Introduction

In recent decades, laser wireless power transmission technology has made significant advancements [1–5]. An increasing number of researchers have become involved in this emerging technology. It eliminates the need for dragging long cables; therefore, laser wireless power transmission (WPT) technology offers a more convenient, comfortable, and enriched lifestyle for humanity [6–10]. With the continuous development and maturation of laser wireless power transmission technology, it will become fully integrated into people's lives, providing power supply for personal electronic devices, wearable/implantable electronic

devices, household appliances, and electric vehicles, among other devices [11–16]. The traditional photovoltaic receiver consists of numerous small-sized photovoltaic cells that are spliced into flat panels, forming photovoltaic panels [17]. When the incident light uniformly illuminates the photovoltaic cells, each cell generates an identical voltage and current. Under these conditions, there will be no circuit loss when all the photovoltaic cells are connected in series or parallel. However, if a photovoltaic cell is damaged, causing it to generate lower current and voltage compared to other cells, it will function as a load and consume power within the circuit. Researchers such as Tiefeng He have found that when the intensity distribution of the incident light is non-uniform, each photovoltaic cell outputs varying voltage and current. Under the same input power conditions, the more non-uniform the light distribution, the lower the output power. This results in significant circuit losses in the system's output circuit, ultimately reducing the overall conversion efficiency of the system [18–22].

In order to address the issue of non-uniform laser irradiation mentioned above, some studies propose modifications to the receiver structure. For instance, researchers such as Shi Dele et al. propose the use of photovoltaic cells of different sizes based on laser intensity [23]. Kumar N et al. introduce a serrated receiver design to reduce reflected light [24]. U. Ortabasi et al. propose a powersphere receiver to enhance uniformity [25,26]. The powersphere is a closed spherical space structure composed of many photovoltaic cells. The closed structure ensures that the reflected laser light will not escape from the powersphere, thereby generating multiple reflections in the powersphere to achieve the effect of light integration, thereby make the illuminance of each point in the cavity is equal. Additionally, some studies also explore algorithms such as I-V curve approximation, Diving rectangle, and a 0.8 Voc-mod model to obtain maximum power point output [27]. Furthermore, improving system conversion efficiency can be achieved by altering circuit structures, including commonly used methods such as SP, TCT, and BL structures [28,29], as well as reconfigurable photovoltaic arrays and dynamic photovoltaic arrays.

From practical effects, none of the above methods alone can achieve the best effect, and there are some shortcomings. For example, due to the high absorption rate of photovoltaic cells, the powersphere cannot achieve the ideal uniform effect in the actual use process. Therefore, exploring the comprehensive use of multiple methods will be the best approach to improve transmission efficiency. For example, on the basis of powersphere, reducing circuit losses through dynamic photovoltaic arrays can further improve transmission efficiency. To effectively utilize a combination of methods, it is necessary to have a comprehensive understanding of the energy distribution variations during laser transmission and analyze the factors influencing energy distribution. This provides a research direction for adjusting laser energy distribution during the transmission process. Thus, this paper establishes a laser wireless power transfer model based on powersphere, which analyzes the variations in laser energy at different stages of transmission through simulation, and analyzes the factors that affect energy distribution. An experimental platform is set up, primarily consisting of a laser, beam expander, Fresnel lens, and powersphere, to conduct experimental validation and lay the foundation for further research.

## 2. Analysis and Experiment of Laser Energy Distribution

### 2.1. System Modeling

The laser WPT system is mainly composed of a launching end, a transmission medium, a receiving end, and load, as shown in Figure 1. The launching end includes a laser power supply, a laser, and a launcher. The transmission medium mainly refers to the substances through which the laser transmits energy, such as air and water. The receiving end includes a harvester, a powersphere, and a circuit control.

The laser power supply provides the electric power. The laser converts electrical energy into light energy. The higher the electro-optical conversion efficiency is, the more light energy can be transmitted under the same input power, then the higher the final power obtained. Due to the high transmission efficiency of 808 nm wavelength lasers in

the air and the high conversion efficiency of photovoltaic cells under this wavelength laser irradiation, semiconductor lasers are commonly used as light sources in laser wireless transmission systems. Considering the generality of the analysis problem, a semiconductor laser with a beam diameter of 400 µm and a divergence angle of 25 degrees is selected as the model light source.

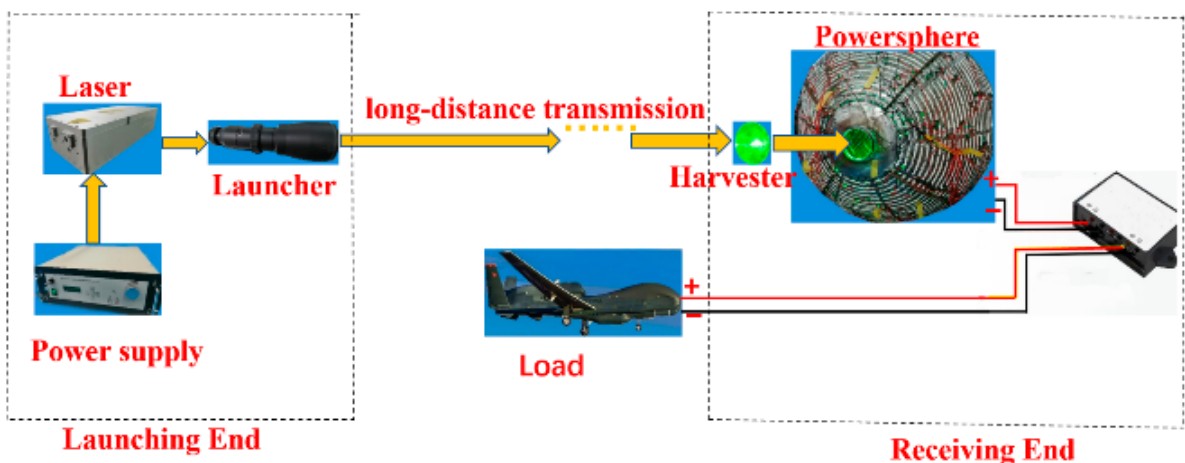

**Figure 1.** System components.

The launcher collimates the transmitted laser and reduces the divergence angle and transmission loss of the laser. The launcher usually has one negative input lens and one positive output lens. The two lenses focus the laser to a common point on the front focus plane of the input lens. Since the beam waist and divergence angle of the laser are a fixed value, when the laser passes through the launcher, the beam waist is larger due to the longer focal length of the output lens, and the divergence angle becomes smaller. Therefore, during the transmission process, the air area for absorbing laser energy is reduced, thereby reducing energy loss. The commonly used launcher is a beam expander. The beam expansion ratio of the beam expander can be selected in accordance with the divergence angle of the laser, the spot diameter of the receiving surface, and the transmission distance. Considering that the transmission distance is small in the simulation and experiment process, the greater the magnification of the beam expander, the more optical lenses are needed, the higher the optical design requirements, and the higher the cost, and the 3× beam expander is currently the simplest and most economical choice. Therefore, a beam expander with a beam expansion rate of three times was selected for system modeling.

Although the divergence angle of the expanded laser becomes smaller, after long-distance transmission, the spot diameter at the receiving end will still be much larger. Since the power before and after transmission is equal, the power density of the enlarged spot will be reduced, especially the power density at the edge of the spot is so low that it is not easy to observe. As a result, this part of the power is ignored and does not enter the powersphere in the aiming process between laser and powersphere, which will cause power loss and reduce transmission efficiency.

To address this issue, it is necessary to use a larger area focusing lens to collect the scattered energy and inject it into the powersphere. A Fresnel lens is composed of numerous concentric circular patterns and can converge incoming parallel light, acting as a convex lens. By retaining only the curvature on the surface and removing as much optical material as possible, a Fresnel lens can be made thin and can have larger dimensions compared to regular lenses. Typically, the diameter can range from 50 mm to 500 mm. Using such a large-aperture Fresnel lens as an energy collector enables the collection of energy distributed over a larger range, thereby improving the efficiency of energy transmission.

The powersphere is an enclosed spherical structure, which is spliced by multiple photovoltaic cells. When the laser enters the powersphere through an incident hole and

irradiates the inner wall of the powersphere, some of the laser is absorbed, while the rest is reflected to other locations on the inner wall of the powersphere. Absorption and reflection occur again, and so on; the luminous flux of each reflected light is the previous luminous flux multiplied by the absorption rate, and the total luminous flux of reflected light is shown in Equation (1) [26]:

$$E_{\sum} = E_1 + E_2 + E_3 + \ldots + E_n = \rho E_B + \rho^2 E_B + \rho^3 E_B + \ldots + \rho^n E_B = \frac{\rho(1-\rho^n)}{1-\rho} E_B \quad (1)$$

It can be seen from the above equation that only when the number of reflections is sufficient, $\rho^n \approx 0$, and the luminous flux at any point on the inner wall is equal. Otherwise, due to the difference in the number of reflections, the right side of the equation is not equal, and the total luminous flux is not equal. Therefore, the use of powersphere instead of photovoltaic panels can improve the light uniformity of the receiving surface.

To enhance the uniformity of light in the powersphere and minimize the errors caused by internal device, powerspheres are typically designed to be relatively large. In the model, a powersphere with a diameter of 1000 mm is used, which is composed of single-crystal silicon solar cells measuring 20 mm × 20 mm. The aperture diameter of the optical system is set to 100 mm. The circuit control mainly involves the photovoltaic array configuration of the output circuit, the maximum power point tracking (MPPT) and group control array technology, which is used to obtain the maximum power output and improve the output power of the system. Using the aforementioned parameters of the semiconductor laser, beam expander, Fresnel lens, and powersphere, a laser wireless power transmission model with the powersphere as the receiver has established through Lighttools, shown in Figure 2. The laser transmission distance is 2000 mm in the transmission model.

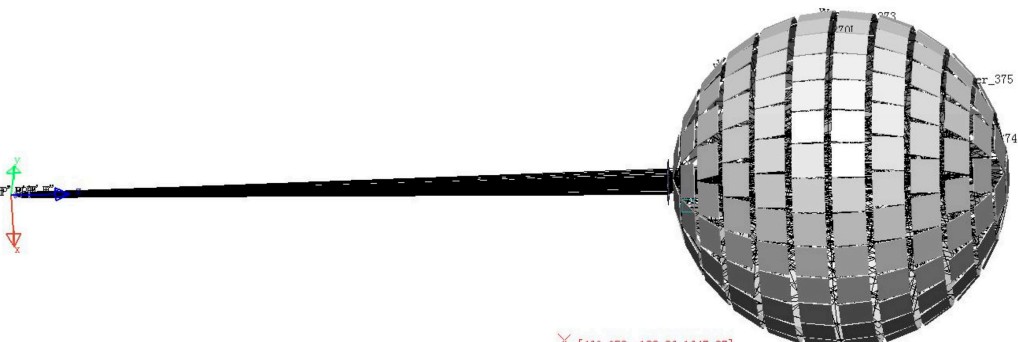

**Figure 2.** Diagram of system model.

### 2.2. Simulation Analysis

Illuminance refers to the luminous flux per unit area projected onto a plane, while intensity refers to the light flux emitted in a unit solid angle, considering that laser only exhibits a spherical distribution when it reaches the surface of a powersphere, and in other stages it has a planar distribution. Therefore, in the process of analyzing the energy distribution, a spherical far-field receiver is only added at the location of the powersphere to reflect the laser energy distribution using the light intensity distribution. Surface receivers are used at other positions to analyze the laser energy distribution based on the illuminance distribution.

After creating the semiconductor laser model with the parameters of a spot diameter of 0.40 mm and a divergence angle of 25° is created using Lighttools, a surface receiver is added at 1.00 mm away from the laser output end for optical tracing. The tracing result is shown in Figure 3. The spot diameter gradually increases, that is, the laser has a divergence angle.

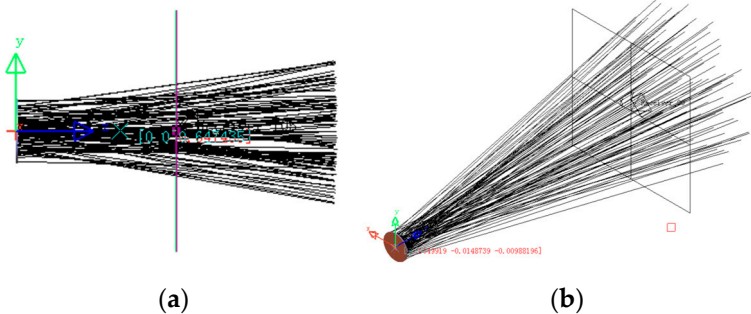

**Figure 3.** Result of laser tracking: (**a**) Front view; (**b**) Side view.

Another surface receiver was added at 2.00 mm to calculate the divergence angle of the laser. The corresponding spot diameters can be obtained from the illumination distribution at the two locations, as shown in Figure 4. Figure 4a is the illuminance distribution at 1.00 mm. From the distribution diagram, the spot radius at 1.00 mm is 0.45 mm, which is larger than the laser output spot radius (0.20 mm). The spot presents an energy distribution with a high center and a low edge. The energy (80%) is mostly concentrated at the spot center, which is a typical Gaussian distribution. This result shows that the model in the software is correct, and the output light of the model is laser. Figure 4b is the illuminance distribution at 2.00 mm. From the distribution diagram, the spot radius at 2.00 mm is 0.68 mm. The spot radius here is 1.5 times that at 1.00 mm. In accordance with the definition of divergence angle, the half angle of divergence is

$$\alpha = arctg\frac{R_2 - R_1}{\Delta L} = arctg\frac{0.68 - 45}{2 - 1} = 12.95° \tag{2}$$

where $R_2$ is the radius of the spot at 2.00 mm, $R_1$ is the radius of the spot at 1.00 mm, and $\Delta L$ is the distance between the two spots.

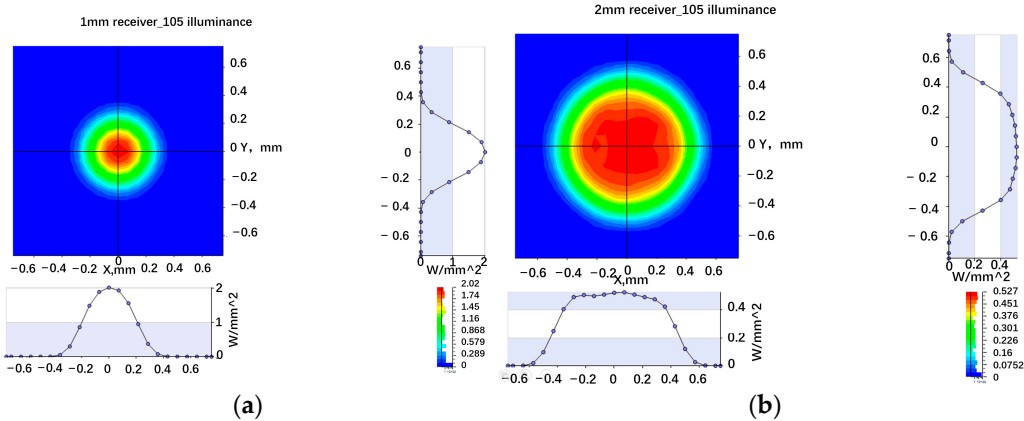

**Figure 4.** Illuminance distribution: (**a**) 1.00 mm; (**b**) 2.00 mm.

Formula (2) shows that the full angle of divergence is 25.9°, which is basically close to the input value of 25°. In consideration of the reading error, the divergence angle in the simulation is equal to the input value; that is, the established model is consistent with the input parameters. Figure 4 indicates that the maximum power density at 1.00 mm is 2.02 W/mm$^2$, and the maximum power density at 2.00 mm is reduced to 0.527 W/mm$^2$. That is, the difference between the maximum and minimum light intensities in the spot decreases, but the area where the power density is large at the spot center is increased. However, the overall spot at 2.00 mm still presents a Gaussian distribution with a strong center and weak edges.

For long-distance optical transmission, excessive divergence angle is the main reason for laser transmission loss. To reduce energy loss, the divergence angle of the laser should

be reduced, which can be achieved using a beam expander for collimation. The beam expander reduces the divergence angle by increasing the beam waist, so that the laser spot will not increase all the time during transmission, reducing the energy absorbed by the air, and increasing the transmission distance and output power of the overall system. With reference to the parameters of the beam expander in the optical manual, a four-piece structure is selected as the design basis. The beam expansion ratio is 3. The parameters of the original beam expander are shown in Table 1. After setting evaluation functions such as divergence angle, beam expansion ratio, and total system length, and using the optimization function of Lighttools software to perform certain optimization, the parameters of the beam expander are shown in Table 2. The beam expander is added to the model for light tracing, and the result is shown in Figure 5. The divergence angle after beam expansion is evidently smaller, and it looks like parallel light from the side of the output light.

**Table 1.** Original Parameters.

| Surface | r (mm) | d (mm) | n |
| --- | --- | --- | --- |
| 1 | 28.45 | 3.79 | 1.4970 |
| 2 | −373.80 | 0.03 | 1.4970 |
| 3 | 27.31 | 3.08 | 1.4970 |
| 4 | 162.76 | 1.88 | 1.4970 |
| 5 | −335.17 | 1.64 | 1.8061 |
| 6 | 27.03 | 0.03 | 1.8061 |
| 7 | 14.07 | 3.79 | 1.4970 |
| 8 | 46.93 | Infinite | 1.4970 |

**Table 2.** Optimized Parameters.

| Surface | r (mm) | d (mm) | n |
| --- | --- | --- | --- |
| 1 | 28.45 | 3.79 | 1.4970 |
| 2 | −373.80 | 0.03 | 1.4970 |
| 3 | 27.31 | 1.64 | 1.4970 |
| 4 | 162.76 | 14.23 | 1.4970 |
| 5 | −335.17 | 3.08 | 1.8061 |
| 6 | 27.03 | 21.12 | 1.4970 |
| 7 | 14.07 | 3.79 | 1.4970 |
| 8 | 46.93 | Infinite | 1.4970 |

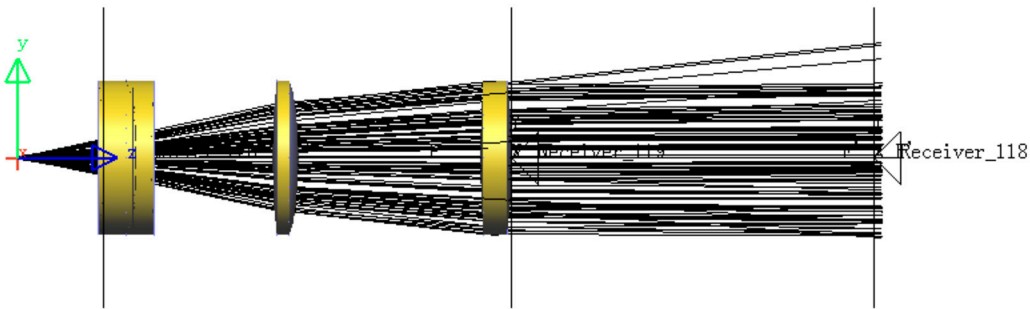

**Figure 5.** Tracking results after beam expansion.

The surface receivers are added at the incident end of the beam expander (10.06 mm), the output end of the beam expander (57.75 mm), and at 200.00 mm. The illuminance distribution of the three positions is shown in Figure 6. Figure 6a presents the illuminance distribution at the incident end of the beam expander, and its spot radius is approximately 5.00 mm. Figure 6b shows the illuminance distribution at the output end of the beam expander, and its spot radius is approximately 15.00 mm. Relative to the spot at the input end, the spot at the output end is magnified by three times. Accordingly, the beam

expansion ratio is 3, which is equal to the beam expansion ratio of the initial structure. Figure 6c depicts the illuminance distribution at 200.00 mm, and the spot radius is also close to 15.00 mm. The divergence angle of the laser is significantly reduced, and the output laser of the beam expander is close to parallel light, which clearly shows that the beam expander used in the model is effective.

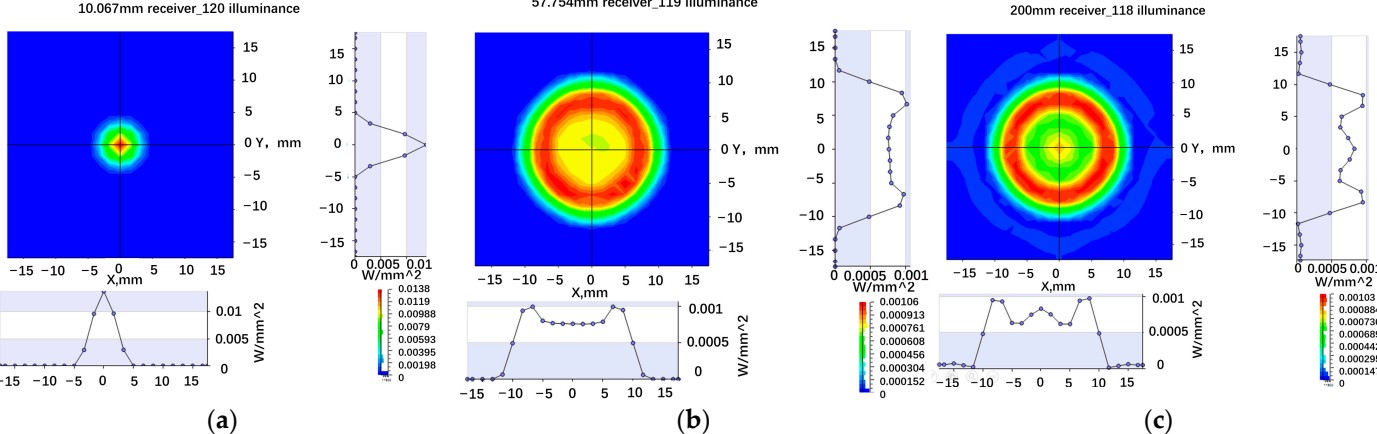

**Figure 6.** Illuminance distribution before and after beam expansion: (**a**) 10.06 mm; (**b**) 57.75 mm; (**c**) 200.00 mm.

From the power density of the three positions in Figure 6, the maximum power density before beam expansion is 0.01380 W/mm$^2$, whereas the maximum power density after beam expansion is 0.00106 W/mm$^2$. That is, the maximum power density after beam expansion decreases to 1/10 before beam expansion, but the area of the center with a high power density is greatly increased. When the laser is transmitted to the 200.00 mm position, the maximum power density becomes 0.00103 W/mm$^2$, which is basically the same as the maximum power density after beam expansion. This result indicates that the laser transmission loss after beam expansion is minimal, which proves that the divergence angle of the laser becomes smaller after beam expansion.

After beam expansion, although the divergence angle of the laser becomes smaller, there still remains a small angle, and the spot will still become large after long-distance transmission. A surface receiver is added at 1980.00 mm to understand the energy distribution of the spot at the receiving end. The illuminance distribution at this position is shown in Figure 7. The maximum power density at 1980.00 mm is 0.00017 W/mm$^2$, which is less than 0.00103 W/mm$^2$ at 200.00 mm. The spot radius also increases from 15.00 mm to 50.00 mm, which is basically equal to the entrance radius of the powersphere. When the transmission distance is longer, the spot radius will be greater than 50.00 mm, and some lasers cannot enter the powersphere. Therefore, a large-aperture focusing lens is needed to reduce the spot radius at the entrance. With its large size and light weight, a Fresnel lens is especially suitable for application in this place.

Under the condition that the incident laser light is perpendicular to the incident surface of the lens, the entrance aperture is placed between the lens and the focus position, through changing the position of the lens and the entrance aperture can change the spot radius at the entrance. Figure 8 is the result of optical tracing after inserting a Fresnel lens with a focal length of 100.00 mm and a size of 120 mm × 120 mm at 1980.00 mm and adding a surface receiver near the focal point. The Fresnel lens focuses the incident laser light at 2021.00 mm and reduces the spot radius at the entrance end of the powersphere, avoiding the energy loss caused by the mismatch between the spot and entrance sizes. From the spot

radius before focusing and the focal length of the lens, the half angle of divergence after focusing can be calculated as follows:

$$\alpha = arctg\frac{R}{F} = arctg\frac{50}{40} = 51.34° \tag{3}$$

where $R$ is the spot radius before focusing, and $F$ is the focal length of the lens.

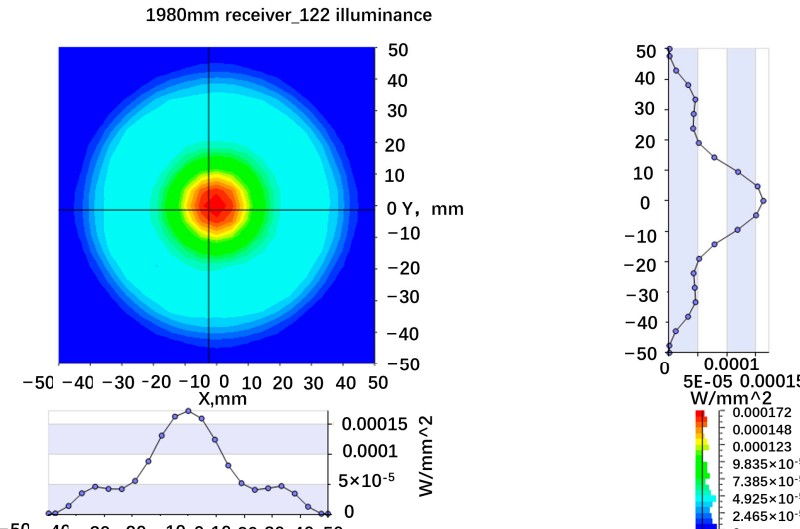

**Figure 7.** Illuminance distribution at 1980.00 mm.

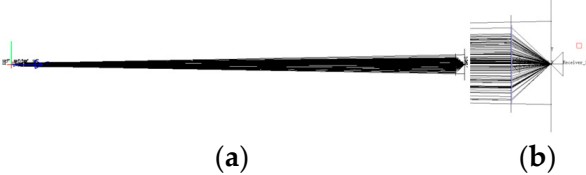

**Figure 8.** Tracking results after focusing by the Fresnel lens: (**a**) Overall tracking results; (**b**) Zoom in on the focused part.

Formula (3) presents that the divergence angle after focusing is related to the focal length of the lens. The smaller the focal length is, the larger the divergence angle will be. The divergence angle after focusing is much larger than the divergence angle at the output end of the laser. The focused laser will directly irradiate the inner surface of the powersphere. Hence, the larger the divergence angle is, the larger the spot and the smaller the maximum power density will be, which are conducive to the light uniform effect of the powersphere. The use of a Fresnel lens with a short focal length will improve the uniformity of the laser at the receiving end. Figure 9 is the illuminance distribution of the laser at the focal point. The spot radius is reduced to 13.00 mm, and its maximum power density is increased to 0.00131 W/mm$^2$, which is close to the maximum power density before beam expansion.

The focused laser will irradiate the photovoltaic cell on the inner wall of the powersphere after entering the powersphere. Part of the laser is absorbed by the photovoltaic cell, and another part is reflected by the photovoltaic cell and transmitted to other photovoltaic cells on the inner wall. This process is repeated until the energy is completely absorbed. A spherical far-field receiver is inserted onto the inner surface of the powersphere for optical tracking. The spherical far-field receiver is a spherical receiver with a radius slightly smaller than the radius of the powersphere. The tracking results are shown in Figure 10. Approximately 30% of the photovoltaic cells in the powersphere can be directly radiated

by the laser. The light density in the indirect irradiation area is less than that in the directly irradiated area. In particular, the area close to the edge of the directly irradiated area has the lowest optical density; that is, the light intensity in this area is the smallest.

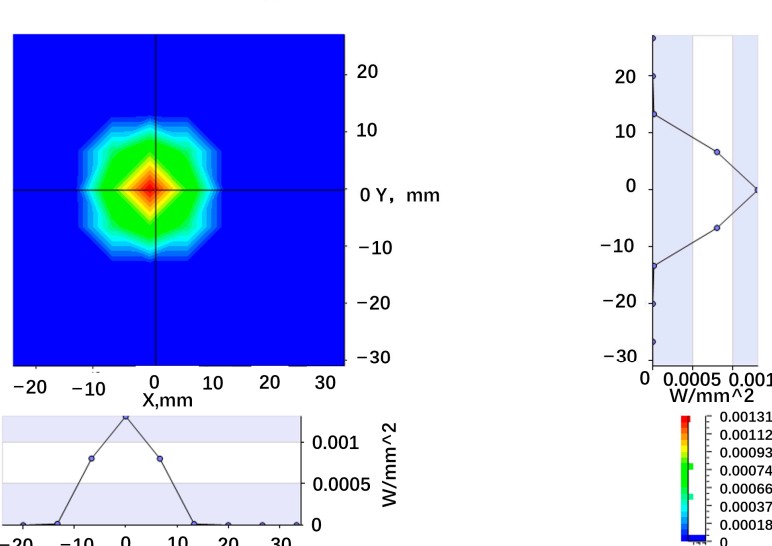

**Figure 9.** Illuminance distribution at the focal.

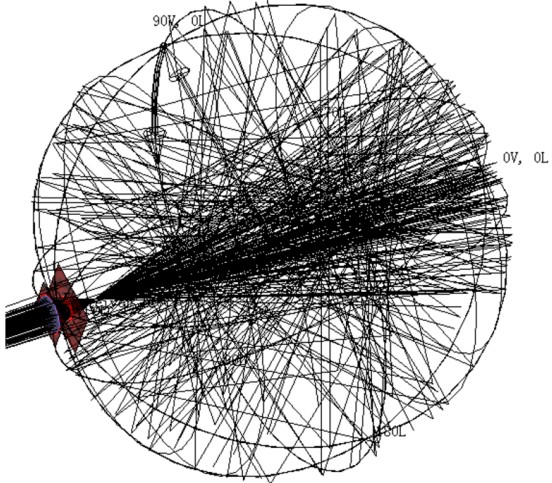

**Figure 10.** Tracking results in the powersphere.

Although the laser transmission can be understood intuitively from Figure 10, the distribution of light intensity in the powersphere is impossible to understand. In order to gain a deeper understanding of the laser wireless power transmission principle, a spherical far-field receiver is used to observe the intensity distribution of the light on the inner surface of the powersphere. The result is shown in Figure 11. Figure 11a is a schematic of the LV coordinate definition of the far-field receiver. The X-axis, Z-axis, and the sphere in the figure form a circular plane. The ray in the plane that starts from the center of the circle and points to the shell of the sphere is latitude, which is the V-axis. The angle range of the V-axis is 0–180°, the positive axis of the Z-axis is 0°, and the negative axis of the Z-axis is 180°, and the rotation direction is counterclockwise. The X-axis, Y-axis, and sphere in the figure form a circular plane. The ray in the plane that starts from the circle center and points to the sphere shell is the longitude, which is the L axis. The angle range of the L axis

is 0–360°, and the positive axis of the X-axis is 0°. The ray returns to the positive X-axis after the longitude is rotated through 360°, and the rotation direction is counterclockwise.

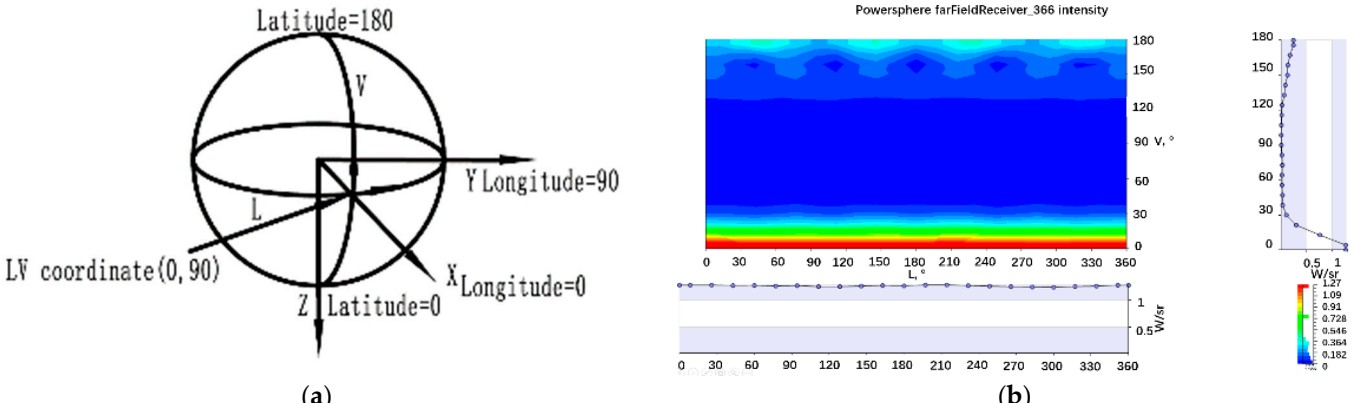

**Figure 11.** Light intensity distribution on the powersphere: (**a**) LV coordinate definition; (**b**) Light intensity distribution.

In accordance with the definition of LV coordinates, the powersphere can be expanded into a rectangular light intensity distribution shown in Figure 11b. In the expanded figure, the V-axis is parallel to the optical axis of the laser, 0° of the V-axis is located at the center point of the end face of the powersphere directly irradiated by the laser, and 180° of the V-axis is located at the center point of the end face where the laser is incident. The L axis is perpendicular to the optical axis of the laser. Since the laser spot is circular, the intensity of light at the same latitude is the same, regardless of its position distribution in coordinates.

Figure 11b demonstrates that the light intensity fluctuates slightly in the range of 0–360° on the L axis, which is consistent with the previous analysis that the area with the same spot diameter has equal light intensity. However, the light intensity will change within the range of 0–180° on the V-axis, and the light intensity change curve can be divided into three parts. The first part of the light intensity gradually decreases as the angle increases until it approaches 0. That is, the light intensity is the strongest at first. The light intensity of the second part is close to 0, but the fluctuations do not change much. The third part of the light intensity increases as the angle increases. The strongest area accounts for about 30% of the total area. This result is basically consistent with that of optical tracing. From Figure 11b, the color in the range of 0–10° in the V coordinate is red. In accordance with the example diagram, the light intensity in this area is the highest, and the highest light intensity can reach 1.27 W/sr. This result means that the area is located at the center of the spot directly irradiated by the laser. As the angle increases, the color changes from red to yellow and then to green. The minimum light intensity in the green area is 0.40 W/sr, which is located at approximately 40° on the V-axis. As the angle further increases, the color changes from green to blue until some areas are close to 0. The angle of these area is 40–120°. The light illuminating this region primarily consists of the low-intensity direct illumination from the edges of the light spot.

After 120°, the color changes from blue to green slowly, and the light intensity gradually increases again. The maximum light intensity reaches approximately 0.30 W/sr, which is located near 180°. Due to the symmetrical distribution of the 140–180° region and the 0–40° region on both sides of the center of the circle, as well as the symmetrical trend in the distribution of light intensity, it indicates that the light illuminating this area is laser light that has undergone first reflection.

The above analysis shows that although the powersphere can form multiple reflections, a large difference in the power of direct and reflected light exists due to the high absorption rate of photovoltaic cells and other reasons. The larger the number of reflections is, the lower the laser power will be, such that the directly irradiated area must be increased to improve the uniformity in the powersphere.

### 2.3. Experiment and Analysis

To verify the results of simulation, an 808 nm semiconductor laser, a beam expander, a Fresnel lens, and a powersphere receiver are used to build a laser WPT experimental platform, as shown in Figure 12. By conducting experiments, it is possible to measure and observe whether the laser power distribution at various stages aligns with the simulations and analyses. This provides a basis for subsequent research of the system.

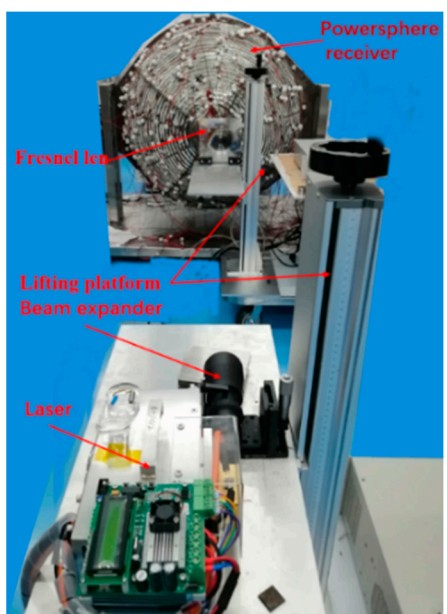

**Figure 12.** Laser WPT experimental platform. Laser WPT experimental platform.

In the experiment, the maximum output power of the 808 nm laser is 50 W, the spot diameter is 0.81 mm, and the full divergence angle is 16°. The output laser spot at 200 mm is shown in Figure 13a. The energy at the spot center is strong, whereas that at the edge is weak. A beam quality analyzer is used to measure the laser spot to understand the energy distribution in the spot, as shown in Figure 13b. Figure 13c,d are the intensity distribution curves of the spot in the vertical and horizontal directions in the beam quality analyzer, respectively. From Figure 13b–d, the laser has a Gaussian distribution with a strong central light intensity and a weak edge light intensity. Due to the different divergence angles of the long and short axes of semiconductor lasers, the energy distribution curves of the beam are inconsistent in the horizontal and vertical directions. Compared with the simulation results, the light intensity distribution curves in Figure 13c,d are not standard Gaussian distributions. This kind of error is mainly caused by the measurement error, the measurement position, and the quality of the semiconductor laser itself. Thus, the simulation and experiment are consistent.

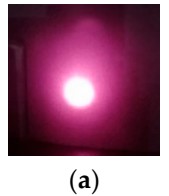
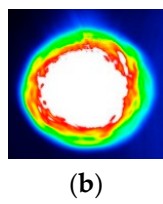
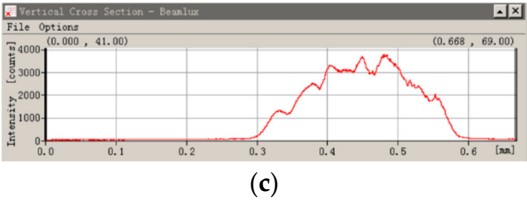
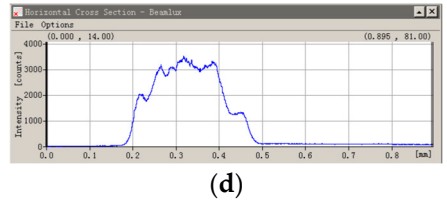

(**a**)  (**b**)  (**c**)  (**d**)

**Figure 13.** Power distribution of laser before beam expansion: (**a**) Spot at 200.00 mm; (**b**) Spot in the beam quality analyzer; (**c**) Distribution curves in the vertical direct; (**d**) Distribution curves in the horizontal direction.

In order to measure the actual divergence angle of the laser, the spot diameter and laser power at 2.00 mm and 200.00 mm positions are measured in the experiment, as shown in Table 3. The laser is transmitted from the 2.00 mm position to the 200.00 mm position, and the spot diameter increases from 2.00 m to 43.50 mm. In accordance with the definition of divergence angle, the half angle of laser divergence is

$$\alpha = arctg\frac{D_2 - D_1}{L_2 - L_1} = arctg\frac{42.5 - 2}{200 - 2} = 11.72° \tag{4}$$

**Table 3.** Spot Diameter and Laser Power at Different Positions.

| Distance (mm) | Spot Diameter (mm) | Laser Power (W) |
|---|---|---|
| 2.00 | 2.00 | 5 |
| 200.00 | 43.50 | 2 |

Then, the full angle of divergence is 23.44°.

The detectable diameter of the laser power meter is only 20.00 mm, but the spot diameter at 200.00 mm is 43.50 mm, which is much larger than the detectable range of the laser power meter. Therefore, the 200.00 mm measurement value of laser power in Table 3 is inaccurate. Taking into account that the spot diameter is twice the detection diameter, and considering that the area beyond the power meter is at the edge of the spot, with significantly lower power compared to the central position, an estimated unmeasured power of 1 W is assumed. Therefore, the total power at 200.00 mm is 3 W, which is 60% of the power at 2.00 mm. This indicates that the laser's divergence is too large, and significant power losses are expected during laser transmission, which is consistent with the simulation results.

To reduce the divergence angle, a 3× beam expander was used in the experiment to collimate the laser output, and the spot after beam expansion is shown in Figure 14a. Compared to Figure 13a, the white area at the center of the collimated spot has decreased, while the pink area has increased. This indicates that after the beam expansion, the area with high intensity in the spot has decreased. To calculate the divergence angle after beam expansion, the spot sizes of two positions are measured, which are located at 250.00 mm and 400.00 mm after the input port of the beam expander. The spot diameters in these positions are 58.00 mm and 68.00 mm. Then the divergence angle is:

$$\alpha = arctg\frac{68 - 58}{400 - 250} = 3.83° \tag{5}$$

From Formula (4), the full angle of divergence after beam expansion is 7.66°, which is much less than that before beam expansion (23.44°). That is, the divergence angle becomes smaller, and the transmission loss will be reduced after beam expansion.

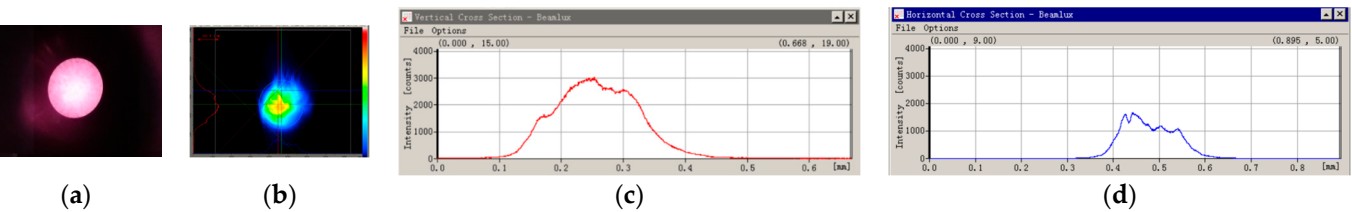

(**a**)      (**b**)      (**c**)      (**d**)

**Figure 14.** Power distribution of laser after beam expansion: (**a**) Spot at 200.00 mm; (**b**) Spot in the beam quality analyzer; (**c**) Distribution curves in the vertical direction; (**d**) Distribution curves in the horizontal direction.

In order to have a more accurate understanding of the intensity distribution within the spot, the beam quality analyzer was used again to measure the spot. The measurement results are shown in Figure 14b. From the graph, it can be observed that the color at the center of the spot has changed from white (before beam expansion) to yellow. The regions with white and red colors in the spot are almost nonexistent, indicating a decrease in power density after the beam expansion. Figure 14c,d show the energy distribution curves of the spot in the vertical and horizontal directions, respectively. Although there is some deformation in the horizontal direction, the energy distribution of the expanded spot still exhibits a Gaussian distribution overall. This deformation will increase the loss of transmission, but the uniformity of light at the receiving end will be improved. But whether the total output will decrease or increase needs to be measured according to specific conditions.

The Gaussian envelope along the long axis of the laser is obtained by fitting the curves of Figures 13c and 14c, as shown in Figure 15a,b. The smooth line in the figure is the envelope line. Comparing the two figures, it can be observed that the central intensity at the spot center after beam expansion is higher than the central intensity at the spot center before beam expansion, but the area of high light intensity is significantly reduced. This means that the beam quality of the laser has improved after beam expansion, resulting in a smaller divergence angle. This observation aligns with the simulation results. Additionally, from Figure 15, it can be seen that the spot diameter has increased from 0.20 mm to 0.30 mm after the beam expansion. This confirms the principle of laser beam expansion, where the spot diameter increases after expansion.

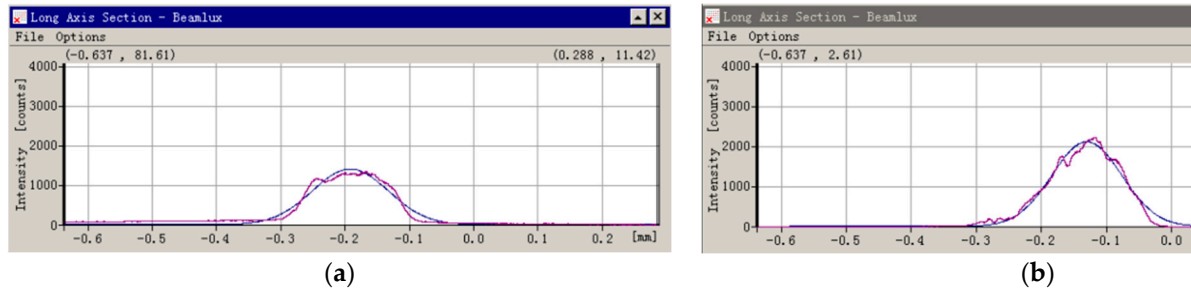

(**a**)                  (**b**)

**Figure 15.** Envelope on the long axis of the laser: (**a**) Before beam expansion; (**b**) After beam expansion.

After long-distance transmission, the laser after expansion reaches the position of 2000.00 mm. The spot of laser at this location is illustrated in Figure 16. Compared to Figure 14a, the energy distribution of the two spots is essentially the same. The main differences are as follows: firstly, the spot size has changed. After a 2000.00 mm transmission, the spot has increased in size from 68 mm to 88 mm. Secondly, there is a change in power. The laser power measured by the power meter at 2000.00 mm is 0.17 W, which is lower than the measured value of 0.45 W at 400.00 mm.

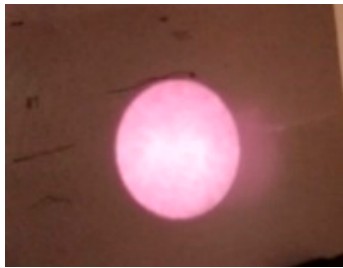

**Figure 16.** Spot at 2000.00 mm.

In order to accurately measure the power of the laser, a Fresnel lens with a focal length of 100.00 mm was used in the experiment to reduce the diameter of the laser spot. The power at that point was then measured using a power meter, and the laser power at a distance of 2000 mm was found to be 0.54 W, which is 3.17 times the power before focusing. This means that focusing the laser using the lens allows the collected laser power to be increased, thereby increasing the power injected into the powersphere. The focused spot is shown in Figure 17a. From the figure, it can be observed that the white region in the center represents a high power density area, while the surrounding pink region represents a low power density area. Figure 17b shows the beam profile measured by the beam quality analyzer. Since the Fresnel lens is composed of numerous concentric circular patterns, the corresponding energy distribution exhibits a large number of concentric circles. Figure 17c,d depict the distribution curves in the vertical and horizontal directions, respectively. From the figures, it can be observed that the laser beam after focusing still exhibits a Gaussian distribution, indicating that it is still non-uniform.

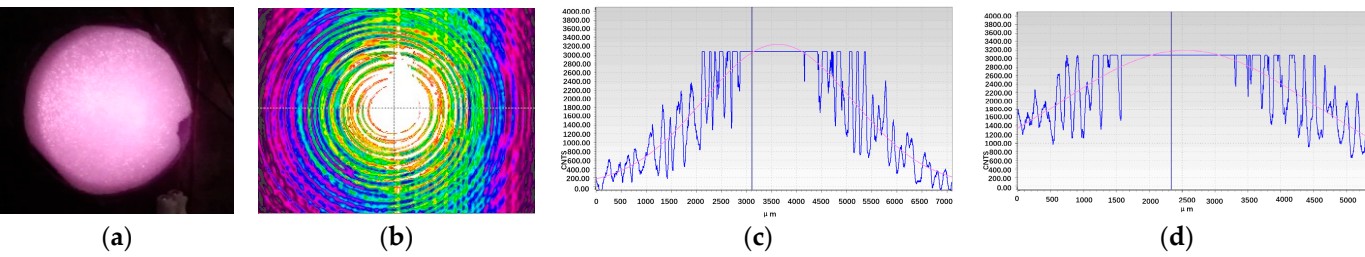

(**a**) (**b**) (**c**) (**d**)

**Figure 17.** Power distribution of laser in a Fresnel lens at a focal length of 100.00 mm: (**a**) Spot; (**b**) Spot in the beam quality analyzer; (**c**) Distribution curves in the vertical direction; (**d**) Distribution curves in the horizontal direction.

After the focused laser enters the powersphere, it undergoes multiple reflections and is completely absorbed by the photovoltaic cells on the inner wall, converting it into electrical power. The powersphere used in the experiment is composed of monocrystalline silicon photovoltaic cells, with each cell measuring 20 mm × 20 mm. The diameter of the powersphere is 1000 mm, and it consists of over 6900 photovoltaic cells, as shown in Figure 2. If all the photovoltaic cells are connected together and there is a short circuit or any other issues at any position, it would result in no output from the entire circuit, making troubleshooting difficult. Therefore, in the experiment, the photovoltaic cells in the powersphere are divided into 69 groups, with each group consisting of 100 cells connected in series. The connection sequence of the cells is shown in Figure 18. From Figure 18, it can be seen that the connection starts from any cell at the top of the powersphere, facing the incident laser direction, and the adjacent cells in the same diameter circle are connected in series in a counterclockwise direction. After connecting the photovoltaic cells on the same diameter, the connection is extended to the photovoltaic cells in an adjacent diameter circle until a series connection of 100 cells is completed. In accordance with this method, all the cells are connected in series in 69 groups.

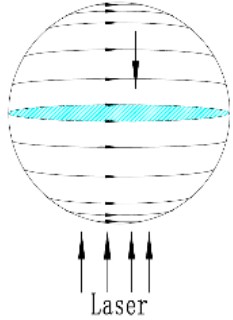

**Figure 18.** Circuit connection.

The powersphere is a closed sphere, and its diameter is extremely large; consequently, it cannot be measured using a power meter or beam quality analyzer. However, the output voltage and current of the photovoltaic cells are proportional to the luminous flux; that is, the output voltage and current are proportional to the light intensity. Therefore, the distribution of laser on the inner surface of the powersphere can be understood by measuring the output voltage and current of the photovoltaic cells at various positions in the powersphere. In the experiment, the voltage and current of the 69 groups of circuits in the powersphere are measured, and the distribution curve is plotted using MATLAB in accordance with the measured data.

The distribution curve of the circuit when the focal length of the Fresnel lens is 100 mm and the distance between the lens and powersphere is 150.00 mm is shown in Figure 19. Figure 19a is the voltage distribution curve, and Figure 19b is the current distribution curve. The abscissa in the figure is not only the serial number of the circuit, but also the V coordinate of the LV coordinate in Figure 11a. In the range of 0–60° of the V coordinate, as the angle increases, the voltage and current gradually decrease. The main reason is that this area is directly irradiated by the laser, and its light intensity is higher than the light intensity in other areas. The smaller the angle is, the closer the position is to the spot center, and the higher the light intensity is. In the range of 60–180° of the V coordinate, the distribution curve basically presents a linear distribution; that is, the light intensity is basically equal.

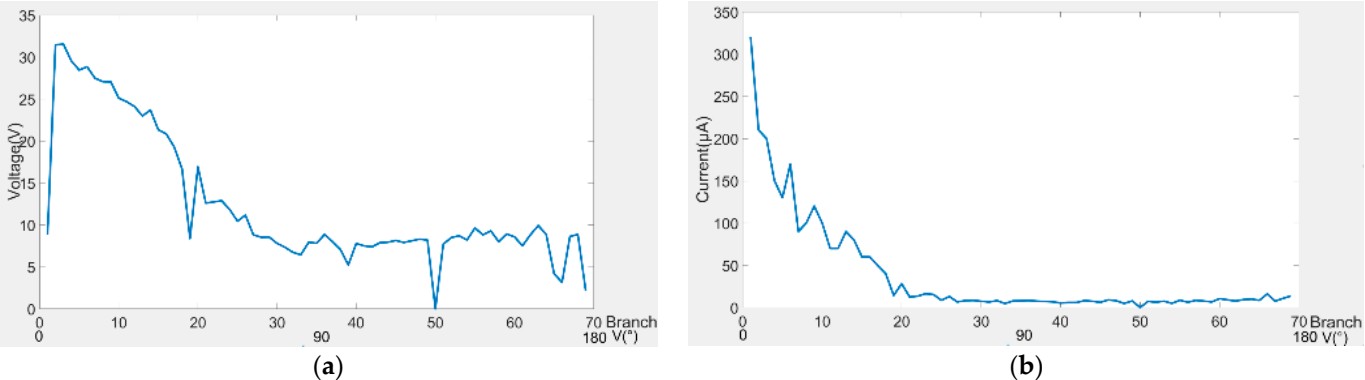

**(a)**                **(b)**

**Figure 19.** Output curve of the powersphere with a Fresnel lens at a focal length of 100.00 mm: (**a**) Voltage curve; (**b**) Current curve.

Due to defects in the manufacturing process of the powersphere, a small number of photovoltaic cells near 0° and 180°, as well as in branches 19 and 50, are unable to output power properly. In order to ensure the overall output is not affected, these cells are disconnected from the circuit during the experiment. As a result, the measured voltage and current in the circuits near 0° and 180°, as well as in branches 19 and 50, will be lower than the actual light intensity. This is also the reason why the measured data near 0° are not at the maximum value. Since each branch is connected in parallel to form the system output, the total power output of the system is equal to the sum of the power of each branch, so the manufacturing defect of the powersphere reduces the output power of the system and should be avoided as much as possible. Comparing the voltage and current curves in Figure 19 with the intensity distribution on the V-axis in Figure 11b, apart from the locations with defects, the three distribution curves are generally consistent. Therefore, the experimental results are in line with the simulated results. From the figures, it can also be observed that the variation in current is greater than the variation in voltage. This is primarily because voltage has a logarithmic relationship with laser radiation intensity, while current has a direct proportionality with laser radiation intensity. As a result, the curve of output voltage changes more gradually compared to the current curve. This is consistent with the principles of photovoltaic conversion.

### 3. Conclusions

The existing laser wireless power transfer systems suffer from low conversion efficiency and low output power. In order to develop high-efficiency laser wireless power transfer technology, it is necessary to understand the intensity distribution of the laser at various stages of wireless energy transmission, in order to propose targeted improvement methods. This paper establishes a laser wireless power transfer model based on a powersphere using Lighttools software, and conducts simulation analysis of the laser transmission process. By analyzing various factors that affect the intensity distribution, the trends of these factors are understood, providing references for further research.

To validate the simulation results, an experimental platform is built using components such as a semiconductor laser, beam expander, Fresnel lens, and powersphere. A beam quality analyzer is used to measure and analyze the laser energy distribution of each component except for the powersphere. The output voltage and current values of each region are measured using a multimeter, reflecting the energy distribution of the powersphere based on the linear relationship between photoelectric current/voltage and light intensity. Simulation and experimentation have shown that the beam expander can reduce the divergence angle, increase the transmission distance, reduce energy loss during transmission, and improve the system's transmission efficiency. The use of a large-aperture focusing mirror in front of the powersphere can increase the collection efficiency of energy and thereby enhance the laser power injected into the powersphere. The focal length of the focusing mirror affects the energy distribution at the receiving end, with a smaller focal length leading to improved uniformity of light and increased output power. This provides a research basis for further enhancing the efficiency of laser wireless power transfer.

**Author Contributions:** T.H. proposed the idea and conceptualization, conducted the simulations, and performed the experiment; Z.L., G.Z. and Q.W. performed scientific discussions and supervised the work; H.H., L.W., K.X. and T.S. helped with revision and organization of the paper. Z.L. and G.Z. also supported funding acquisition. All authors have read and agreed to the published version of the manuscript.

**Funding:** This research was funded by Natural Science Foundation of Top Talent of SZTU (GDRC202108), Guangdong Provincial Major Scientific Research Grant (Grant No. 2022KQNCX073) and Shenzhen Science and Technology Program (Grant No. 202207191410140001).

**Institutional Review Board Statement:** Not applicable.

**Informed Consent Statement:** Not applicable.

**Data Availability Statement:** Not applicable.

**Conflicts of Interest:** The authors declare no conflict of interest.

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
