# Peer review of "Analysis and Experiment of Laser Energy Distribution of Laser Wireless Power Transmission Based on a Powersphere Receiver"

_photonics, doi:10.3390/photonics10070844_

Round 1

Reviewer 1 Report

This paper aims to develop high-efficiency laser wireless power transfer technology by understanding the intensity distribution of the laser at different stages of wireless energy transmission. Through simulation and experimentation, it is shown that using a beam expander can reduce divergence angle and energy loss, while employing a large-aperture focusing mirror enhances energy collection and output power, providing a basis for improving the efficiency of laser wireless power transfer. These suggestions have the potential to enhance the quality of your paper:

1-    You should add the significant findings, and the conclusions reached to your abstract.

Introduction:

2-    The statement that none of the individual methods can achieve optimal results and the suggestion to explore the comprehensive use of multiple methods is thought-provoking. It would be helpful to expand on why the combination of methods is necessary and what potential synergies or benefits it could offer.

3-    It would be useful to briefly explain what powersphere is and its role in addressing the issue of non-uniform laser irradiation.

Analysis and Experiment of Laser Energy Distribution:

4-    Provide a brief explanation of how the launcher reduces the divergence angle and transmission loss of the laser.

5-    Mention any specific technical considerations or constraints that led to the selection of a beam expander with a magnification of 3x.

6-    Provide a brief explanation of how the powersphere traps the incident laser and ensures equal light intensity on the spherical surface.

7-    Clarify the role of circuit control and briefly mentioning its purpose.

8-    Discuss the impact of the beam expander on the overall system performance.

9-    How were the optimized parameters obtained, and were any specific criteria or performance metrics considered in the optimization?

10- Did you consider any specific measurement techniques or calculations to determine the illuminance distribution?

11- Explain how the lens is incorporated into the system and its impact on reducing the spot radius at the entrance.

12- Providing some insights into the limitations and potential sources of error in the experimental setup would be beneficial.

13- Since the energy distribution curves in the experiment deviate from standard Gaussian distributions, it would be valuable to discuss the potential implications of this deviation.

14- Are there any specific trends or patterns observed in the experimental results that were not evident in the simulation?

Moderate editing of English language required

Reviewer 2 Report

The authors use the optical software Lighttools to study a laser power transmission model based on powersphere. The energy distribution changes of the propagated laser power are studied through various components. An experimental verification is done using a 50 W  808 nm laser, a beam expander, Fresnel lens, and large powersphere. The study is relevant for the field of optical wireless power transmission to help understanding the energy distribution variations during laser transmission and to analyze the factors affecting the transmitted light. The study shows that a Fresnel lens can be used as an energy collector to improve the collection of energy distributed over a larger range. The study is for a laser transmission distance of 2000mm. The powersphere has a diameter of 1m and is composed of single crystal silicon solar cells of 20mm×20mm. The aperture diameter of the optical system is 100mm.

The study reveals that the the voltage and current curves with the intensity distribution from the model are generally consistent with the experimental results. Also, simulation and experimentation have shown that the beam expander can reduce the divergence angle, increase the transmission distance, reduce energy loss during transmission, and improve the transmission efficiency. The use of a large‐aperture focusing mirror in front of the powersphere was shown to increase the collection efficiency.

I think the paper is of interested for publication. The following revisions are recommended:

Include units in table 1 and table 2 (at least in the caption if not for each columns), and any other similar omissions.

The authors should verify and adjust the number of digit accuracy? 

For example 57.754mm, is it really precise to a micron…?

In addition, I’m not sure I understand why the incident light is supposed to be reflected so many times, I thought most of the light would be absorbed upon incidence with the first cell it reaches, and only a very small fraction would make it to more than 1 or 2 reflections. The authors should explain this topic better and be more quantitative on their statement.

The related descriptions was not convincing for me: “The powersphere is a closed spherical space structure composed of many photovoltaic cells. It can “trap” the incident laser in a relatively closed cavity, such that the unabsorbed laser can reach every part of the sphere surface after multiple reflections. In this way, the light intensity of each part of the spherical surface is equal, which solves the problem of nonuniform distribution of laser intensity on the photovoltaic receiving surface.” 

Verify and correct typo errors: ‘aera’ for example, etc

Some figures have small fonts or could be clearer.

The paper is long, so perhaps cutting some less critical parts could improve the manuscript.

Round 2

Reviewer 1 Report

Some of the figures need to be edited. The quality of some of them is very low, such as Figures 1 and 5.  The texts inside the photos are generally unclear and unreadable. The plots screened from Windows need to be replaced ( Figures 13-15).

Moderate editing of the English language is required.

Reviewer 2 Report

The authors addressed the initial main concerns, I think the paper is now adequate for publication.